# Outcome Measures of In-Office Endovenous Radiofrequency Treatment of Varicose Vein Feasibility

**DOI:** 10.3390/diagnostics13020327

**Published:** 2023-01-16

**Authors:** Alfonso Pannone, Alessia Di Girolamo, Matteo Orrico, Nicola Mangialardi

**Affiliations:** 1Department of Vascular Surgery, Azienda Ospedaliera San Camillo-Forlanini, Circonvallazione Gianicolense 87, 00152 Rome, Italy; 2Policlinico Umberto I, Sapienza University of Rome, Viale del Policlinico 155, 00185 Rome, Italy

**Keywords:** radiofrequency ablation, varicose veins, deep vein thrombosis

## Abstract

RFA is a relatively new treatment, approved by the FDA in 1999 and is a minimally invasive intervention that has become one of the most common alternatives due to its many advantages, including decreased pain, less morbidity, shorter hospital stay and faster return to work. We retrospectively analyzed a total of 503 limbs submitted for surgical interventions for VVs using the RFA, combined or not with surgical phlebectomies and sclerotherapy, in our institution between April 2012 and December 2020. The technical success was 99.8%, as in one case the RFA catheter arrested at the first third of the thigh due to the tortuosity of the vessel. On the first post-operative day, the mean VAS scale was 0.32 ± 0.56 (0–2). Perioperative complications occurred in 77 cases (15.3%): intraoperative pain in 24 cases, in nine cases associated with vagal syndrome, superficial hematoma in 30 cases, EHIT in seven cases, phlebitis in 14 cases and paresthesia in two cases. RFA procedures had been shown to be cost-effective therapeutic options in adult patients requiring treatment due to the incompetence of the GSV. In our study, we highlighted that this technique is feasible as an ambulatory procedure; it improves quality of life and symptoms in the majority of patients with varicose veins, with a rapid return to normal life and work activities.

## 1. Introduction

The malfunction of the venous system, including the superficial or deep lower limbs’ venous system, can lead to venous insufficiency [1].

The prevalence of venous disease is very high among adults, ranging from 40% to 80% and seems to be highest in the Western world [2]. In fact, in western countries, an estimated 23% of adults have varicose veins and 6% have more advanced chronic venous disease, including all stages from skin changes to healed or active venous ulcers [3].

Chronic venous insufficiency can result in significant morbidity due to chronic pain that leads to loss of workdays [4].

Treatment of venous insufficiency includes both non-invasive and invasive methods. Among non-invasive treatments are medical therapy and the elasto-compressive stocking.

Among invasive treatment of varicose veins (VVs), the standard treatment in recent decades has been surgery involving high ligation and stripping of the saphenous veins.

However, minimally invasive techniques, i.e., the endovenous techniques, including endovenous laser ablation (EVLA) and endovenous radiofrequency ablation (RFA), have been introduced some years ago and since then have been widely applied.

In particular, RFA was approved by the FDA in 1999 and has become one of the most common alternatives, due to its advantages of decreased pain, less morbidity, shorter hospital stay and faster return to work. Catheters, used for the radiofrequency treatment, are built with a heating element tip that uses thermal energy to destroy the endothelium of the vein, inducing the complete sclerosis of the vein submitted to the treatment [4,5].

The aim of our study was to evaluate the safety and efficacy of RFA in varicose veins treatment in a single center experience.

## 2. Materials and Methods

This retrospective study was approved by the institutional review board and written informed consent was obtained from all of the patients.

A total of 503 limbs underwent surgical interventions for VVs using RFA, combined or not with surgical phlebectomies and sclerotherapy, in our institution between April 2012 and December 2020.

At the time of the diagnosis, demographic features, clinical features, predisposing family history and past personal history were analyzed.

Preoperative ultrasound assessed great saphenous vein incompetence, small saphenous vein incompetence, perforating vein incompetence, deep venous insufficiency, the diameter of the great saphenous and the distance between veins and skin, at three different level of the lower limb, to estimate the possibility of post-operative skin hyperpigmentation and to assess if endovascular occlusion was the most appropriate approach.

The severity of venous insufficiency was categorized according to the Clinical Etiology Anatomy Pathophysiology (CEAP) classification, while the presence of pain was assessed using the VAS scale in the preoperative and post-operative period, as well as during the follow-up clinical controls.

Through a 18 F needle under ultrasound guidance, access to the refluxing great saphenous vein is obtained. In general, the insertion point is limited to 15 distal to the knee joint because the leg GSV is usually continent and is more superficial than the thigh GSV and to avoid saphenic nerve heating damage.

The radiofrequency ablation catheter is then advanced under ultrasound guidance and placed at least 2 cm distal to the saphenofemoral junction, avoiding the epigastric vein confluence. After catheter positioning, a tumescent anesthetic solution is injected with a peristaltic pump around the vein under ultrasound guidance along the entire course of the vein, in order to insulate the surrounding soft tissue, nerves and deep vessels from heat injury and help to compress the target vein, increasing contact of the heating element with the vein walls.

All procedures were performed using the VeinCLEAR™ System Catheter, with 7 cm ablation in each segment, with two cycles of 20 second treatment. The RF generator is then activated, which results in segmental heat energy of 120 degrees Celsius being applied.

When necessary, RFA treatment was combined with phlebectomy and perforator ligation. After all the necessary phlebectomies, the residual varicosities were assessed and adjunct sclerotherapy was performed if the diameter was more than 5 mm. Both the phlebectomies and the sclerotherapy were preferably deferred until 3 months after the RFA procedure, as many varicosities diminish in size over time post-procedure.

Sclerotherapy was performed mixing saline solution and air, using 3 mL syringes connected three-way. The solution was prepared by to and fro flow of the agents carried out at least 20 times and the final solution was injected in the residual varicosities.

All procedures were performed under local anesthesia in outpatient surgery; in case of anxious patients, a mild sedation with Midazolam was performed.

A tumescent anesthesia mixture, with 1 adrenaline vial, 1 lidocaine 2% vial and 1 ropivacaine 7.5% vial in 500 mL of cold saline solution, was administered in all patients.

Low Molecular Weight Heparin was advised in all patients, except for those in anticoagulant therapy for other reasons. Compression therapy was administered after the ablation procedure. In case of enormous leg varicosities, an elastic compression bandage was applied for 5–7 days followed by a Class II (23–32 mmHg at the ankle) compression stocking for 1 month. Ibuprofen 600 mg was prescribed and patients were advised to take it if they experienced pain.

After day-surgery treatment, the patients were asked to return for a follow-up visit one week after the procedure and pain was evaluated using the VAS scale. Every patient was advised to return back to working activities as soon as they felt well enough.

All patients were enrolled in a follow-up protocol, including DUS at 1 month, 6 months, 12 and 24 months.

The presence of varicosities within 1 month was classified as residual varicosities and most of the patients were addressed to adjunctive treatment.

The recurrence or the new appearance of varicosities at the 6 months follow-up control were considered as recurrent varicosities.

The clinical success was considered as absence of recurrent or persistent varicosities at 6, 12 and 24 months. The persistent varicosities at 1 month were not considered as a parameter for clinical success, considering that most of cases were staged procedures. (Figure 1 and Figure 2).

Statistical analysis was conducted using the 26th version of IBM SPSS statistics. The Pearson chi-square test was used with nominal variables, while ANOVA univariate analysis was used with ordinal continuous variables.

## 3. Results

The preoperative patients’ characteristics are summarized in Table 1. The mean age was 53.29 ± 11.98 years (20–82 years) and 336 (67%) were female. A total of 501 (99.6%) limbs had VVs of the great saphenous vein, while 2 (0.4%) had VVs of the small saphenous vein. The CEAP classification before RFA was: C1 in 42 cases (8.34%), C2 in 294 cases (58.45%), C3 in 113 cases (22.46%), C4a in 19 cases (3.7%), C4b in 10 cases (1.98%), C5 in 10 cases (1.98%) and C6 in 15 cases (2.98%) (Table 2).

In the clinical history, Deep Vein Thrombosis (DVT) was reported in nine patients (1.78%), Superficial Vein Thrombosis in 89 patients (17.67%), while 81 patients (16.1%) underwent previous venous surgery, with stripping of the great saphenous vein in two cases (0.4%). The mean Venous Clinical Severity Score (VCSS) before RFA was 6.38 ± 3.6 (0–22), while mean preoperative VAS scale was 2.61 ± 1.98 (0–8).

The target vessel of the RFA treatment was the great saphenous vein in 486 cases (96.6%) (right in 224 and left in 262) and the anterior saphenous vein in 17 cases (3.38%) (right in 7, left in 10).

The preoperative ultrasound assessed the great saphenous vein incompetence in 500 cases (99.4%), small saphenous vein incompetence in three cases (0.59%), perforating vein incompetence in 215 cases (42.74%) and deep venous insufficiency in 34 cases (6.7%).

The mean sapheno-femoral junction (SFJ) diameter was 9.28 ± 3.01 mm (6–20), the mean GSV diameter measured at the proximal third was 8.98 ± 2.95 mm (5–20), the mean GSV diameter measured at the mid third was 8.71 ± 2.66 mm (4–20) and the mean GSV diameter measured at the distal third was 7.52 ± 2.15 mm (3–13).

The mean distance between the skin and the SFJ was 6.47 ± 1.43 mm (4–11), mean distance between the skin and the GSV measured at the proximal third was 8.97 ± 4.95 mm (5–15), measured at the mid third was 8.85 ± 2.23 mm (4–19) and measured at the distal third was 7.99 ± 2.01 mm (3–19). The mean lesion length was 32.76 ± 5.96 mm (15–48).

The RFA treatment was performed under local anesthesia in all cases (100%), while in 37 cases (7.35%) additional sedation was needed.

In 128 cases (25.45%), an associated sclerotherapy treatment was performed with different timing: in 13 cases in the same intervention, in 115 cases 3 months later and in one case 12 months later.

In 248 cases (49.3%), an associated phlebectomy treatment was performed with different timing: in 15 cases in the same intervention, in 232 cases 3 month later and in one case 12 months later. The phlebectomy was performed at the leg in 195 cases and at the thigh in 53 cases.

The RFA catheter length was 60 cm in 123 cases (24.45%) and 100 cm in 380 cases (75.54%).

The mean procedure time was 37.6 ± 10.29 min (13–90). LMWH was administered in 499 cases and elastic compression therapy in all patients.

The technical success was 99.8%, as in one case the RFA catheter arrested at the first third of the thigh due to the tortuosity of the vessel. On first post-operative day, the mean VAS scale was 0.32 ± 0.56 (0–2).

Perioperative complications occurred in 77 cases (15.3%): intraoperative pain in 24 cases, in nine cases associated with vagal syndrome; superficial hematoma in 30 cases, EHIT in seven cases, phlebitis in 14 cases and paresthesia in two cases Table 3.

EHIT presentation and resolution are shown respectively in Figure 3 and Figure 4.

Intraoperative pain was assessed by asking if during the procedure the patient had felt pain. The patients who answered affirmatively were included. In case of recurrent pain during the procedure, additional sedation was performed.

The presence of superficial ecchymosis or hematoma, noted at the end of the procedure, was mainly due to local anesthesia with tumescence and, in the first period of this experience, also due to the learning curve regarding venous catheterization.

No patients needed hospitalization and no major vessel injuries and bleeding were recorded.

Clinical control and Duplex Ultrasound were performed at 1 month in 494 patients, 6 months in 489 patients and 12 months in 472 patients. At 24 months, only 338 patients showed up at follow-up controls.

At 1 months, recanalization of the treated vein was observed in six patients (in two cases from the SFJ for 5 cm), phlebitis occurred in 12 and DVT in 0, hyperpigmentation in nine patients, EHIT in two, matting in 14 and paresthesia in 15. Residual Varicosities were observed in 334 patients, in 36 cases occurring both in the leg and in the thigh, in two cases in the thigh, in 271 in the leg, while in 25 this was not specified. The CEAP classification 1 month after RFA was: C0 in 21 cases (4.25%), C1 in 124 cases (25.10%), C2 in 262 cases (53.04%), C3 in 45 cases (9.11%), C4a in 18 cases (3.64%), C4b in nine cases (1.82%), C5 in 16 cases (3.23%) and C6 in eight cases (1.62%). The mean Venous Clinical Severity Score (VCSS) 1 month after RFA was 3.7 ± 2.49 (0–18) (Table 4).

At 6 months, recanalization was observed in eight patients, hyperpigmentation in two, EHIT in two, matting in six and paresthesia in 11. Recurrent VVs were observed in 56 patients. The clinical success was 88.54%. The CEAP classification 6 months after RFA was: C0 in 128 cases (25.7%), C1 in 259 cases (52.96%), C2 in 43 cases (8.79%), C3 in 12 cases (2.45%), C4a in 19 cases (3.88%), C4b in 11 cases (2.25%), C5 in 17 cases (3.47%) and C6 in 0 cases. The mean Venous Clinical Severity Score (VCSS) 6 months after RFA was 2.4 ± 1.94 (0–8).

At 12 months, recanalization was observed in nine patients, hyperpigmentation in six, paresthesia in one. Recurrent VVs were observed in 57 patients. The clinical success was 87.92%. The CEAP classification 12 months after RFA was: C0 in 143 cases (30.49%), C1 in 240 cases (51.17%), C2 in 35 cases (7.46%), C3 in seven cases (1.49%), C4a in 21 cases (4.48%), C4b in 16 cases (3.41%), C5 in seven cases (1.49%) and C6 in 0 cases. The mean Venous Clinical Severity Score (VCSS) 12 months after RFA was 1.76 ± 1.73 (0–10).

At the last follow-up (24–36 months), recanalization was observed in 12 patients and hyperpigmentation in one patient. Varicosities was observed in 75 patients: in 43 patients (8.55%) there were recurrent varicosities, while in 32 patients (6.36%) there were persistent varicosities. The recurrent varicosities were associated with the recanalization of the treated vessel in five cases (0.99%) and in all other cases with the incontinence of a non-treated vessel (anterior saphenous vein incontinence recorded in six cases). The clinical success was 77.81%.

The CEAP classification 24 months after RFA was: C0 in 68 cases (20.11%), C1 in 185 cases (54.73%), C2 in 48 cases (14.2%), C3 in seven cases (2.07%), C4a in 15 cases (4.43%), C4b in 10 cases (2.96%), C5 in five cases (1.47%) and C6 in 0 cases. The mean Venous Clinical Severity Score (VCSS) 12 months after RFA was 1.82 ± 1.65 (0–8). The follow-up results are summarized in Table 4.

The univariate analysis showed a correlation between the recurrent varicosities and the great saphenous vein diameter, measured at the proximal third (*p*-value 0.001), medium third (*p*-value 0.003) and distal third (*p*-value 0.005), while there was no significant correlation with the sapheno-femoral ostium diameter (*p*-value 0.470).

The univariate analysis, as expected, showed a correlation, despite not always significant, between the matting, the hyperpigmentation and the EHIT with the distance of the vessel from the skin, measured at the sapheno-femoral ostium (SFO), the proximal third, the medium third and the distal third. The *p*-values are showed in Table 5.

The cross-tabs, analyzed with Pearson chi-square test showed no correlation between recanalization and venous disease familiarity, pregnancy, smoking habit, diabetes, dyslipidemia, arterial hypertension, previous DVT, previous thrombophlebitis, or elastic stocking, while the results showed a correlation between recanalization and EBPM administration (*p*-value 0.000).

## 4. Discussion

Chronic venous disease hold fourth place among the chronic disease, with a prevalence of 10–50% in the male adult population and of 50–55% in the female.

Nowadays, treatment options include endovascular (such as RFA and laser therapy), sclerotherapy and surgical therapies.

Compared with standard surgical treatment such as high ligation with stripping of the saphenous veins, the effectiveness and safety of endovascular treatment of VVs have been well demonstrated in a number of studies [6].

Endovenous techniques have been recommended for the treatment of VVs by the Society for Vascular Surgery and the American Venous Forum [3], due to decreased pain, less morbidity, shorter hospital stay and faster return to work with similar efficacy on the patient side. Nevertheless, with the endovenous technique, an operating theatre setting, anesthesiologist assistance or paramedic personal dedicated to the post-operative course, as well as in-patient rooms, are not needed.

In the literature, endovenous techniques are often described as performed on outpatients. Due to this fact and to less pain experienced by the patients, these procedures are usually performed under local anesthesia.

Instead, the high ligation and stripping is usually performed within an operating theatre setting, under spinal or general anesthesia. The need for anesthetic also subjects individuals to further risk of complications (i.e., allergic reaction to anesthetic agents, damage to teeth during intubation, post- operative nausea and vomiting).

The rate of post-procedural complications and peri-operative complications after RFA are very low, with rare thrombotic events, occurring at a similar rate to those after other endovenous and surgical treatments [7].

From the literature analysis, there are four randomized trials comparing the results after RFA with safenectomy, including 200 patients treated, one multicenter clinical registry including 1005 patients, two large monocentric studies including 1322 patients and one single study comparing endovascular laser with RFA.

All the four randomized trials highlight that the patients treated with RFA had significantly less pain, less post-operative morbidity, a better quality of life and a faster return to work, than patients treated by safenectomy.

The results of the Endovascular Vein Occlusion versus Ligation and Vein Stripping Study (EVOLVeS) trial, comparing the early postoperative course after conventional GSV safenectomy and high ligation and after treatment with the RFA using the Closure catheter (VNUS Medical Technologies), has demonstrated the efficacy of the endovascular obliteration of the GSV. The QOL instrument showed advantages in term of discomfort’s amount and cosmetic aspects related to the procedures, defining this procedure as patient friendly.

The retrospective nature of our study did not allow evaluation of the quality of life and of the discomfort of patients after the procedure. However the VAS scale submitted on the first post-operative day confirmed the almost complete absence of post-operative pain.

Visual Analogue Scale (VAS) is recommended for assessment of pain intensity and we decides to use it on the first post-operative day, both for pain and consequent difficulty in returning to normal activity.

The early return to normal activity followed by a rapid return to work bears important considerations for overall cost of the procedure [8].

In terms of cost, articles have been published analysing both the direct and indirect costs of the varicose vein treatment, but the conclusions are not univocal [9].

In fact, despite that the cost seems excessive for a single intervention, compared to conventional surgery it is also clear that the novel technique’s advantages are in terms of indirect costs due to less post-operative comorbidity and recovery time.

The choice of dividing the intervention into two different times, firstly the GSV ablation and secondly the varicose vein sclerotherapy or phlebectomies, gave the advantage of reducing the number and volume of varicosities that need to be treated, and is time and cost sparing.

The absence of major complication such as femoral or other vessels injuries, in addiction to what has been previously described, supports that the technique is a good candidate for in-office surgery.

The presence of residual varicosities should not be misinterpreted as affecting the clinical success, but only the presence of recurrent varicosities at 6 or more months affected this. When comparing the recurrence rate for RFA to HL + Stripping, the rates are similar at three years, but a possible long-term benefit for RFA was seen.

A meta-analysis including all the RCTs of the great saphenous vein treatment with Five Year follow-up was published by Haman et al in 2017, comparing all possible therapies for chronic vein insufficiency, showing the superiority of endovenous techniques and high ligation and GSV stripping, compared to other treatment, in term of success rates [10,11].

Since 2013, international and national guidelines recommended the use of EVTA as the first line treatment intervention for varicose veins [12]. After EVTA treatment is Ultrasound Guided Foam Sclerotherapy (UGFS) and lastly surgical intervention [13,14].

In 2021, a Cochrane Review updated the results of the same study group from 2016. Whing et al. had identified 11 new RCTs, including in the meta-analysis 24 RCTs with 5135 participants. However, there were no new trials for the comparison of RFA and SFJ ligation and stripping of GSV [12].

When considering the DUS examination at the groin in studies comparing endovascular and surgical interventions, the appearance of recurrent reflux after surgery is different from that after endovascular treatment. In fact, neo-vascularization at the SFJ is more frequent after surgery, whereas after EVT residual or newly developed reflux in SFJ tributaries and accessory veins plays a major role. However, according to the literature, after 5 years these different DUS findings did not result in any difference in clinical and QoL outcome [15].

In our series, the recurrence of varicosities in the leg or in the thigh was associated with the recanalization of the target vessel only in less than 1% of cases, underlining the efficacy of the RFA treatment [16].

Nevertheless, this illustrates that even the best treatment methods for patients with GSV incompetence will never result in absolute success in the long-term for all patients, in spite of all efforts to improve treatment strategies, due to the fact that lower limb varicose veins depend on several factors, including increase of body mass index, professional activity, genetic predisposition and pregnancy after treatment. All these factors contribute to the progression or recurrence of disease, so the evolution is not only determined by the technical performance at the moment of intervention, but will also depend on several other interfering factors, such as genetic predisposition, increase of body mass index, professional activity and pregnancy after treatment [17].

Moreover, endothermal closure is usually limited to the proximal segment of GSV, limiting the procedural time and pain and, in perspective, maintaining the best vascular graft at calf level [18].

## 5. Conclusions

RFA procedures have been shown to be cost-effective therapeutic options in adult patients requiring treatment for incompetence of GSV. It is important to underline that this technique is feasible as an ambulatory procedure and improves quality of life and symptoms in the majority of patients with varicose veins, with a rapid return to normal life and work activities.

Despite what was previously thought, after analyzing the literature and our experience, it is as equally effective in reducing long term recurrence and risk of reoperation.

## Figures and Tables

**Figure 1 diagnostics-13-00327-f001:**
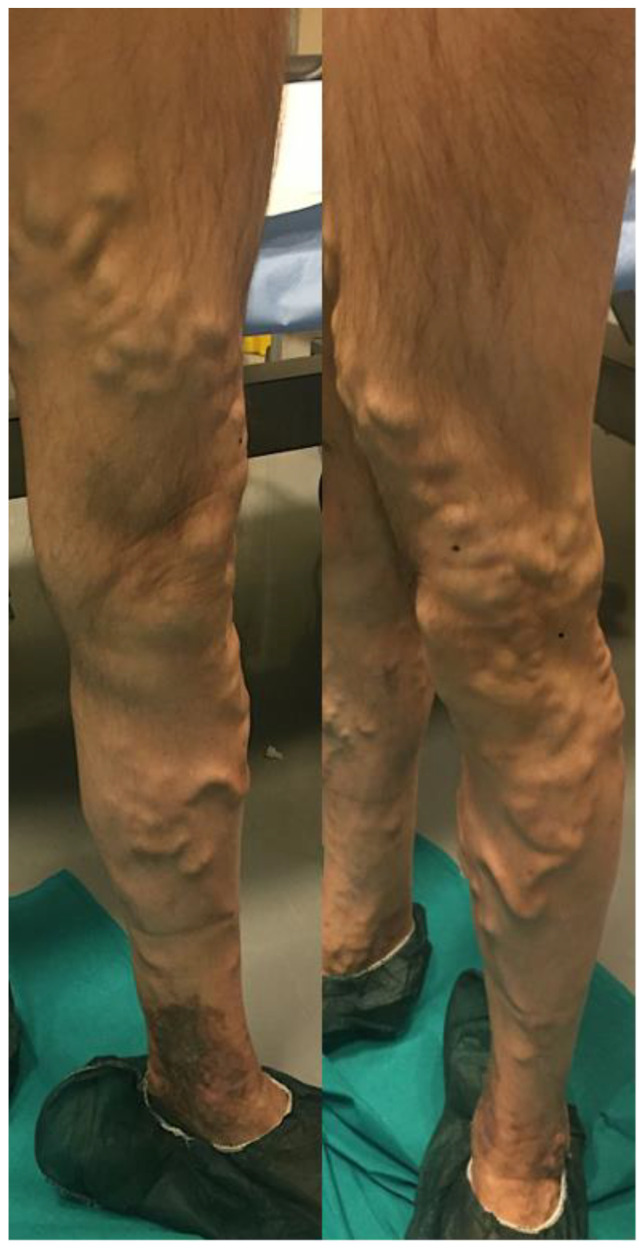
Preoperative photo of varicose veins of the great saphenous vein and collateral veins of the lower left limb.

**Figure 2 diagnostics-13-00327-f002:**
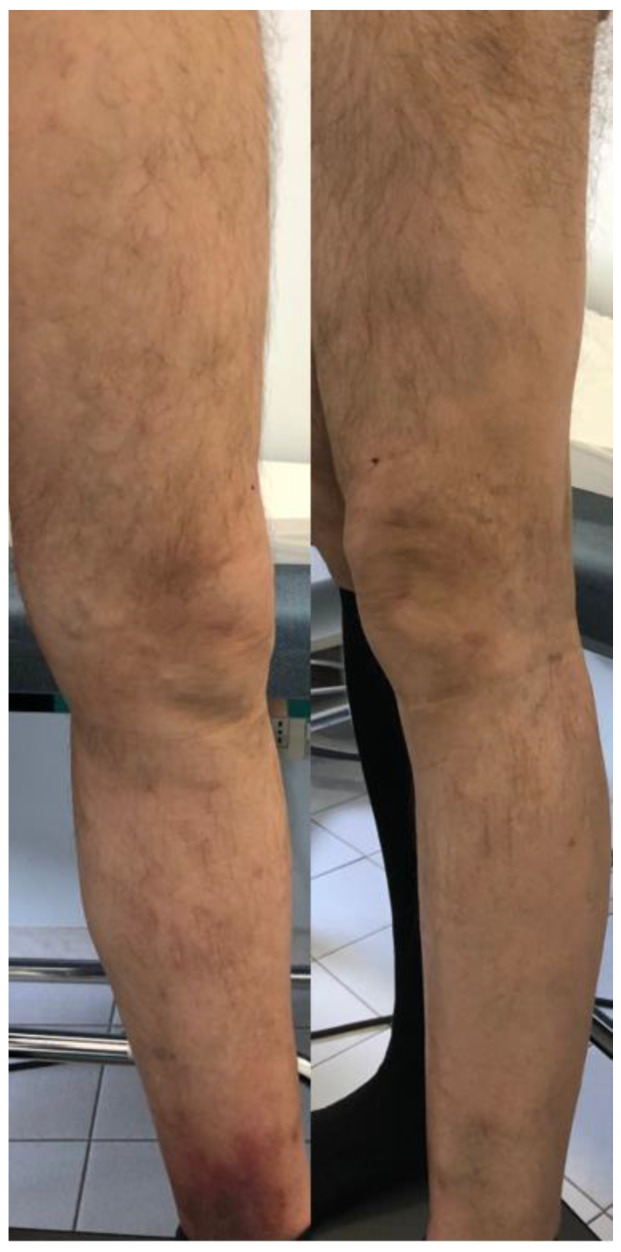
1 year post-operative photo of the lower left limb treated with radiofrequency ablation of the Great saphenous vein, flebectomies and sclerotherapy of the collateral veins with absence of recurrent or persistent varicosities.

**Figure 3 diagnostics-13-00327-f003:**
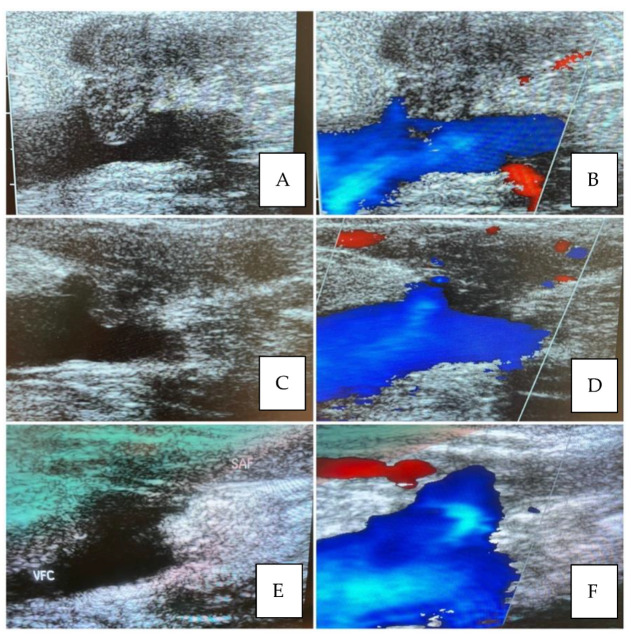
CASE OF EHIT: (**A**,**B**): Great Saphenous Vein Thrombosis reaching the Sapheno-Femoral Junction and projecting in the Common Femoral Vein. This is one case of the nine EHIT occurring after Radiofrequency treatment. (**C**,**D**): Partial Regression Of The Thrombosis After 7 Days And Initial Restored Patency Of The Epigastric Vein. (**E**,**F**): Complete Regression Of The Thrombus And Complete Restored Patency Of The Epigastric Vein.

**Figure 4 diagnostics-13-00327-f004:**
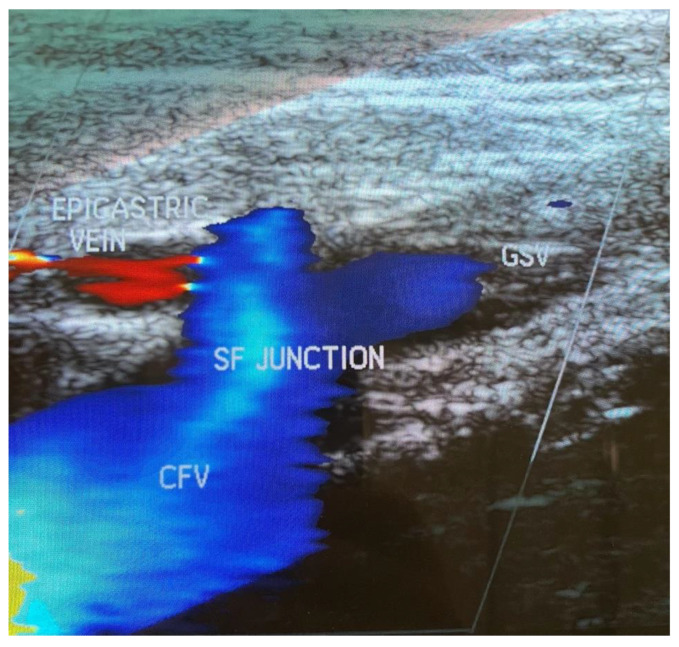
Post-operative obliteration of the Great Saphenous Vein, about 2 cm before the confluence in the SF junction with patency of the epigastric vein.

**Table 1 diagnostics-13-00327-t001:** Preoperative patients’ characteristics.

Age (Year)	53.29
Sex (female)	336 (67%)
Prevalence	
Hypertension	157 (31.21%)
Dyslipidemia	112 (22.26%)
DM	68 (13.51%)
Smoke habit	134 (26.64%)
Thyroid disorders	93 (18.49%)
CAD	32 (6.36%)
Pregnancy	231 (45.9%)
Familiarity	175 (34.8%)
Thrombophilia (MTHFR mutation)	11 (2.19%)

**Table 2 diagnostics-13-00327-t002:** The CEAP classification before RFA.

CEAP Classification	Number of Patients (%)
C1	42 (8.34%)
C2	294 (58.45%)
C3	113 (22.46%)
C4a	19 (3.7%)
C4b	10 (1.98%)
C5	10 (1.98%)
C6	15 (2.98%)

**Table 3 diagnostics-13-00327-t003:** The perioperative complications.

Complications	Number of Patients (%)
Intraoperative Pain	24 (4.77%)
Intraoperative Pain associated with Vagal Syndrome	9 (1.79%)
Superficial Hematoma	30 (5.96%)
EHIT	7 (1.39%)
Phlebitis	14 (2.78%)
Paresthesia	2 (0.39%)

**Table 4 diagnostics-13-00327-t004:** The mean Venous Clinical Severity Score (VCSS).

Complications	1 Month	6 Months	12 Months
Recanalization	6 (1.20%)	8 (1.63%)	9 (1.91%)
Phlebitis	12 (2.40%)	0	0
DVT	2 (0.40%)	0	0
EHIT	2 (0.40%)	6 (1.22%)	0
Paresthesia	15 (3.04%)	11 (2.25%)	1 (0.21%)
Hyperpigmentation	9 (1.82%)	2 (0.41%)	6 (1.27%)
Matting	14 (2.83%)	6 (1.27%)	0
Recurrent VVs	334 (67.61%)	56 (11.45%)	57 (12.07%)

**Table 5 diagnostics-13-00327-t005:** *p*-values.

	SFO	PROX. THIRD	MID THIRD	DIST. THIRD
Hyperpigmentation	0.000	0.901	0.002	0.000
EHIT	0.000	0.002	0.968	1.000
Matting	1.000	0.000	0.965	0.003

## Data Availability

The data presented in this study are available on request from the corresponding author. The data are not publicly available due to privacy law.

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
