# Peer review of "Outcome Measures of In-Office Endovenous Radiofrequency Treatment of Varicose Vein Feasibility"

_diagnostics, 2023, doi:10.3390/diagnostics13020327_

Round 1
Reviewer 1 Report
The article is well structured and deals with a topical issue. The statistical study is well documented, the results are well outlined. The bibliography is well chosen and suggestive, including current references. It is a well-worked article and deserves to be published.
Author Response
Dear reviewer,
We thank you for the revision.
Reviewer 2 Report
1. Line 58 - Why should we measure the diameter and the distance between the vein and the skin at three different levels? What is the significance of these data in clinical surgical indications?
2. Line 65 - What causes the entrance to be limited to 15 cm at the distal end of the knee joint?
3. Line 175 - How to solve the problem of intraoperative and postoperative complications? Will other treatments have similar complications? Have measures been taken for the complications with high incidence?
4. Line 295 - Is there any specific data analysis support for RFA in reducing the risk of long-term recurrence and reoperation?
5. The average lesion length of veins, combined sclerotherapy at different times and other data analyzed in the internal analysis were not reflected in the discussion. Whether these data will affect the clinical efficacy?
Author Response
Dear reviewer,
Thank you for your revision. We have modified the introduction, discussion and conclusion section in order to be more clear in explaining the study.
Line 58 - Why should we measure the diameter and the distance between the vein and the skin at three different levels? What is the significance of these data in clinical surgical indications?
We perform these measurements "to estimate the possibility of post-operative skin hyperpigmentation and to assess if the endovascular occlusion was the most appropriate approach. "
Line 65 - What causes the entrance to be limited to 15 cm at the distal end of the knee joint?
"The insertion point is limited to 15 distal to the knee joint because the leg GSV is usually continent and is more superficial than the thigh GSV and to avoid saphenic nerve heating damage."
Line 175 - How to solve the problem of intraoperative and postoperative complications? Will other treatments have similar complications? Have measures been taken for the complications with high incidence?
In order to answer to your question, we added the following part: "The intraoperative pain has been assessed asking if during the procedure the patient has felt pain. The patients who answered affirmatively were included. In case of recurrent pain during the procedure, an additional sedation was performed. The presence of superficial ecchymosis or hematoma, noted at the end of the procedure, was mainly due mainly to local anesthesia with tumescence and in the first period of this experience, also due to the learning curve in venous catheterization."
Line 295 - Is there any specific data analysis support for RFA in reducing the risk of long-term recurrence and reoperation?
In order to answer your question we have revised the discussion section from line 292-333.
The average lesion length of veins data were not present in our study due to the retrospective nature.
The discussion were revised in order to add data of the combined sclerotherapy at different times and other data analyzed in the internal analysis.
Round 2
Reviewer 2 Report
The author has answered all my questions and I agreed to accept it.